# Myelin Oligodendrocyte Glycoprotein (MOG)35–55 Mannan Conjugate Induces Human T-Cell Tolerance and Can Be Used as a Personalized Therapy for Multiple Sclerosis

**DOI:** 10.3390/ijms25116092

**Published:** 2024-05-31

**Authors:** Maria Rodi, Anne-Lise de Lastic, Ioannis Panagoulias, Ioanna Aggeletopoulou, Kostas Kelaidonis, John Matsoukas, Vasso Apostolopoulos, Athanasia Mouzaki

**Affiliations:** 1Laboratory of Immunohematology, Medical School, University of Patras, 26500 Patras, Greece; marodi_biol@yahoo.gr (M.R.); delastic@gmail.com (A.-L.d.L.); iopanagoulia@upatras.gr (I.P.); iaggel@hotmail.com (I.A.); 2NewDrug P.C., Patras Science Park, 26504 Patras, Greece; k.kelaidonis@gmail.com (K.K.); imats1953@gmail.com (J.M.); 3Immunology and Translational Research, Institute for Health and Sport, Victoria University, Melbourne, VIC 3030, Australia; vasso.apostolopoulos@vu.edu.au; 4Department of Physiology and Pharmacology, Cumming School of Medicine, University of Calgary, Calgary, AB T2N1N4, Canada; 5Department of Chemistry, University of Patras, 26504 Patras, Greece; 6Immunology Program, Australian Institute for Musculoskeletal Science (AIMSS), Melbourne, VIC 3021, Australia

**Keywords:** peptides, MOG35–55, mannan, vitamin D, dendritic cells, cytokines, regulatory T cells, immunomodulation, human

## Abstract

We have previously performed preclinical studies with the oxidized mannan-conjugated peptide MOG35–55 (OM-MOG35–55) in vivo (EAE mouse model) and in vitro (human peripheral blood) and demonstrated that OM-MOG35–55 suppresses antigen-specific T cell responses associated with autoimmune demyelination. Based on these results, we developed different types of dendritic cells (DCs) from the peripheral blood monocytes of patients with multiple sclerosis (MS) or healthy controls presenting OM-MOG35–55 or MOG-35–55 to autologous T cells to investigate the tolerogenic potential of OM-MOG35–55 for its possible use in MS therapy. To this end, monocytes were differentiated into different DC types in the presence of IL-4+GM-CSF ± dexamethasone (DEXA) ± vitamin D3 (VITD3). At the end of their differentiation, the DCs were loaded with peptides and co-cultured with T cells +IL-2 for 4 antigen presentation cycles. The phenotypes of the DC and T cell populations were analyzed using flow cytometry and the secreted cytokines using flow cytometry or ELISA. On day 8, the monocytes had converted into DCs expressing the typical markers of mature or immature phenotypes. Co-culture of T cells with all DC types for 4 antigen presentation cycles resulted in an increase in memory CD4+ T cells compared to memory CD8+ T cells and a suppressive shift in secreted cytokines, mainly due to increased TGF-β1 levels. The best tolerogenic effect was obtained when patient CD4+ T cells were co-cultured with VITD3-DCs presenting OM-MOG35–55, resulting in the highest levels of CD4+PD-1+ T cells and CD4+CD25+Foxp3+ Τ cells. In conclusion, the tolerance induction protocols presented in this work demonstrate that OM-MOG35–55 could form the basis for the development of personalized therapeutic vaccines or immunomodulatory treatments for MS.

## 1. Introduction

MS is a chronic demyelinating disease of the central nervous system (CNS) with an inflammatory and autoimmune etiology. A reduced number or dysfunctional regulatory cells, especially CD4+CD25+Foxp3+ T regulatory cells (Tregs), overactive effector CD4+ helper T (Th) cells, especially Th1 and Th17, CD8+ cytotoxic T cells, autoantibody production and activated antigen-presenting cells (APCs), including dendritic cells (DCs), play a critical role in mediating an inflammatory milieu that leads to an autoimmune attack on intrinsic protein components within the myelin sheath. This leads to axonal damage and neurodegeneration [1]. The best-characterized autoantigens in MS are myelin basic protein (MBP), proteolipid protein (PLP) and myelin oligodendrocyte glycoprotein (MOG) [2].

A promising approach to combat autoimmune diseases such as MS is immunotherapies aimed at restoring tolerance and avoiding the use of non-specific immunosuppressive drugs or biological agents such as monoclonal antibodies. These include cyclic peptides based on MBP, PLP and MOG, as well as altered peptide ligands (APL), which are closely related to native peptides (agonists or wild-type) and have 1–2 substituted amino acid residues that interact with the T cell receptor (TCR) but retain their binding ability to the human leukocyte antigen (HLA) [3,4,5,6].

Human clinical trials (phase I, II or III) conducted with MBP peptides, agonists or APLs yielded unsatisfactory results, either because of a lack of efficacy or because, although they were effective in blocking or switching autoreactive clones, they caused side effects such as the development of immediate-type hypersensitivity reactions, the formation of antibodies that cross-reacted with native MBP or poor tolerability [7,8,9,10]. Therefore, further extensive preclinical testing is required, and new peptides must be used with an appropriate carrier that induces tolerance or alters the resulting immune response.

Mannan, a poly-mannose isolated from the wall of yeast cells, has been shown to bind to the mannose receptor on DCs and is a ligand for Toll-like receptor 4 [11]. Mannan conjugated to the cancer protein MUC1 elicits an immune and protective response in mice, and its translation into human clinical trials has shown both immunologic and clinical efficacy [12,13]. Due to the immunomodulatory properties of mannan, its effect as a carrier for MS peptides is being investigated by our group [14].

Mannan in oxidized or reduced form conjugated to the immunodominant agonist MOG35–55 peptide protected mice in prophylactic and therapeutic protocols against experimental autoimmune encephalomyelitis (EAE, an animal model for MS), with oxidized mannan-conjugated MOG35–55 (OM-MOG35–55) yielding the best results. Protection was peptide-specific and was associated with reduced antigen-specific T cell proliferation but not with changes in Th1, Th17 and T regulatory cell (Treg) differentiation or T cell apoptosis compared to EAE in controls [15]. Furthermore, humanized HLA-DR2 transgenic mice immunized with OM-MOG35–55 were protected against EAE in both prophylactic and therapeutic protocols [16]. 

In a previous study [17], in which peripheral blood from patients with relapsing-remitting MS (RRMS, *n* = 83) and healthy controls (*n* = 45) was used to investigate the types of regulatory cells and how they are affected by disease activity and type of treatment, we were able to show that in patients in the acute phase of the disease without therapy, the concentration of CD4+CD25+Foxp3+ T regulatory cells (Tregs) was significantly reduced compared to healthy controls and that Tregs responded to various peptides mapping to myelin antigens in culture with proliferation and cytokine secretion, with the OM-MOG35–55 peptide having the best tolerogenic effect. In addition, the stability and integrity of OM-MOG35–55 were confirmed using analytical and enzymatic methods [18,19].

To further these studies, we developed different types of DCs from peripheral blood monocytes of MS patients presenting OM-MOG35–55 or MOG-35–55 to autologous T cells to investigate the tolerogenic potential of OM-MOG35–55. Our working hypothesis is that the OM-MOG35–55 conjugate is a strong candidate for a therapeutic vaccine or immunomodulatory treatment of MS in the context of personalized medicine.

## 2. Results

### 2.1. Development of DCs from Peripheral Blood Monocytes

Monocytes isolated from PBMCs of RRMS patients and controls were differentiated into different DC types in the presence of IL-4 and GM-CSF with or without the addition of DEXA or VITD3 [20,21]. At the end of their differentiation, DCs were loaded with OM-MOG35–55 or MOG35–55 and received the LPS maturation signal [22]. They were then co-cultured with autologous, non-adherent PBMCs in the presence of IL-2 for 4 antigen presentation cycles (Figure 1).

Phenotypic analysis of adherent PBMCs on day 0 of culture showed that they were 100% CD14+ monocytes. On day 8 of culture, the cells had the morphology of DCs, as determined using light microscopy. Phenotypic analysis of the cells showed that they expressed all markers typical of DCs, i.e., HLA-DR, CD40, CD80, CD83 and CD86 (Figure 2). DCs generated with VITD3 or VITD3+DEXA showed the most typical semi-mature phenotype with low expression of CD80 [23,24]. The results were similar between patient and control DCs.

### 2.2. Cytokines Secreted by the Different DC Types

DCs secreted the pro-inflammatory (type-1) cytokines IL-1, IL-6, IL-8, IL-12 and TNF-α and the anti-inflammatory (type-2) cytokines IL-10 and TGF-β. Overall, the highest secretion rates were observed for IL-8 and TGF-β. Compared to control-derived DCs, patient-derived CTRL-DCs secreted significantly higher amounts of IL-1, IL-6, IL-8, IL-12 and TNF-α. Patient-derived VITD3-DCs secreted significantly lower amounts of IL-1, IL-10 and IL-12 (Figure 3).

The ratio of type-2/type-1 (anti-inflammatory/pro-inflammatory) cytokines in the different DC cultures, reflecting an effector or suppressor shift in the overall cytokine profiles, shows a strong suppressor shift in the cytokines secreted by patient-derived DCs generated from monocytes in the presence of VITD3 or DEXA. A suppressor shift was not observed in control-derived DCs generated from monocytes in the presence of DEXA, VITD3 or DEXA+VITD3; control-derived VITD3-DCs had the highest suppressor cytokine profile, but the mean difference from the suppressor cytokine profile of CTRL-DCs did not reach statistical significance (Figure 4).

### 2.3. Effect of Antigen Presentation by DCs to Autologous T Cells

#### 2.3.1. Beginning of Cultures

The viability of the stored lymphocytes was estimated to be over 70% before the cells were added to the DC cultures (Appendix A).

Phenotypic analysis of lymphocytes (Figure 5A) showed significant differences in the concentrations of CD4+ T cells and B cells; specifically, patients had a 10% lower concentration of CD4+ T cells and twice the concentration of B cells than the control group. In addition, patients had a significantly lower concentration of naive (CD45RA) CD4+ and CD8+ T cells and a significantly higher concentration of memory (CD45RO) CD4+ and CD8+ T cells than controls (Figure 5B,C).

#### 2.3.2. End of Cultures

Cells

At the end of the culture and after four rounds of antigen presentation, the lymphocytes in the cultures consisted exclusively of T cells (>99.5%). As determined using light microscopy, DCs were still present in the cultures. In the cultures with DEXA+VITD3-DCs, there were very few viable cells that could not be characterized via phenotypic analysis. In the remaining cultures, all T cells were viable. Phenotypic analysis performed on days 33 and 36 of culture showed that the majority of T cells derived from both patients and controls and cultured with the different types of DCs presenting OM-MOG35–55 or MOG35–55 consisted of CD4+ T cells (Figure 6A). T cells cultured with CTRL-DCs presenting MOG35–55 contained the highest proportion of CD8+ T cells (Figure 6B). T cells cultured with CTRL-DCs presenting MOG35–55 or OM-MOG35–55 contained the highest proportion of CD4-CD8-T cells (Figure 6C).

Phenotypic analysis of CD4+ and CD8+ T cells showed that culture with all types of DCs mainly promoted the generation of memory CD4+ T cells derived from both patients and controls (Figure 7A,B), and to a much lower extent, the generation of memory CD8+ T cells (Figure 7C,D).

OM-MOG35–55-specific CD4+ T cells derived from the patients exhibited significantly increased PD-1 expression compared to control-derived CD4+ T cells when cultured for 33 days with DEXA-DCs and VITD3-DCs and on day 36 under all culture conditions, reaching a maximum of over 80% when cultured with VITD3-DCs (Figure 8A). In addition, the proportion of OM-MOG35–55-specific Tregs derived from the patients was significantly higher than that of Tregs from the control group, reaching a maximum of over 30% when cultured with VITD3-DCs on day 36 (Figure 8B).

Cytokines

At the end of the 36-day culture and after four rounds of antigen presentation, the results in the cultures with DCs derived from patients or controls were different in terms of cytokine secretion. In the cultures with DCs presenting OM-MOG35–55, the secreted cytokines were IL-4, IL-6, IL-10, IFN-γ, TNF-α and TGF-β, with TGF-β having the highest concentration; IL-17 was not detected in any of the cultures (Figure 9).

In the cultures of DCs with MOG35–55, the secreted cytokines were IL-6, IL-10, IFN-γ, TNF-α and TGF-β, with TGF-β having the highest concentration; IL-4 and IL-17 were not detected in any of the cultures (Figure 10).

The cultures with the MOG35–55 peptide contained significantly lower TGF-β levels compared to the cultures with OM-MOG35–55. In the cultures of DCs presenting OM-MOG35–55, the strongest suppressor cytokine shift was observed in the cultures with DEXA-DCs, whereas in the cultures of DCs presenting MOG35–55, the strongest suppressor cytokine shift was observed in the cultures with VITD3-DCs (Figure 9 and Figure 10).

## 3. Discussion

Our data suggest that OM-MOG35–55-presenting DCs induce T cell tolerance that is maintained via the administration of free OM-MOG35–55 peptides at 11-day intervals. OM-MOG35–55 peptides induced IL-4 secretion, whereas MOG35–55 did not, and furthermore, OM-MOG35–55 induced much higher TGF-β secretion, suggesting that this was due to mannan. Comparing the results of DC cultures with OM-MOG35–55 or MOG35–55 peptides, it is clear that the type of DCs is the main factor influencing the outcome of MOG35–55 presentation to T cells, while conjugation of the peptide to mannan appears to be the key factor for tolerance induction and overrides the maturation state of DCs. These results confirm our previous finding that the peptide OM-MOG35–55 elicits the best tolerogenic effect when added to PBMC cultures from RRMS patients [17].

Our data also emphasize the importance of dexamethasone and vitamin D in the creation of tolerogenic DCs [20,21,25]. Our results show that vitamin D has a better effect in this regard on DCs derived from healthy individuals. Moreover, the use of vitamin D or dexamethasone alone in the differentiation of monocytes into tolerogenic DCs resulted in the maintenance of viable T cells in long-term cultures compared to the joint use of vitamin D and dexamethasone. This is an interesting point that could be useful in experimental studies on the differentiation of human monocytes into DCs and the use of peptides in vaccination protocols. Importantly, presentation of OM-MOG35–55 by DCs generated in the presence of vitamin D resulted in the highest levels of CD4+PD-1+ and CD4+CD25+Foxp3+ T cells, which are significantly reduced in MS patients [17,26]. Considering that carefully monitored vitamin D supplementation has been shown to improve MS and other inflammatory conditions [27,28,29], the administration of OM-MOG35–55 to patients together with vitamin D supplementation should also be considered.

The antigen presentation system developed by our research team to test the induction of tolerance of host T cells to MS antigens needs to be thoroughly evaluated with a larger number of peripheral blood samples from MS patients to determine the phenotype of the resulting T cell clones, their TCR repertoire, their function, their proliferation potential, their phenotypic stability over time and over several cycles of antigen challenge, and finally their suppressive potential against autologous effector T cells isolated from fresh blood samples from the same patients. In addition, peripheral blood samples from patients with RRMS with relatively low EDSS were used in this study. It would be interesting to test the effect of OM-MOG35–55 in other forms of MS and a broader EDSS spectrum.

A prerequisite for the clinical testing of a potential vaccine or immunomodulatory treatment with conjugated peptides in humans is the efficacy and stability of the peptides and conjugates. We have already conducted preclinical and clinical studies with mannan conjugated to a cancer protein and demonstrated its efficacy and safety [12,13]. We have also performed preclinical studies with mannan-conjugated peptides mapping to myelin epitopes in the EAE mouse model [14,15,16] and in human peripheral blood [17] and demonstrated that oxidized mannan gave the best results, suggesting that OM-peptides may be useful for suppressing antigen-specific CD4+ T cell responses associated with autoimmune CNS demyelination. We have also demonstrated the stability and integrity of OM-MOG35–55 [18,19].

More and more studies are looking at the manipulation of the immune system as a means of controlling or curing various diseases, from malignancies to autoimmune diseases. Harnessing the body’s own processes of antigen presentation is a promising tactic with a limited number of side effects and a broad spectrum of activity [24,30].

Our group is working on the development of peptide-based treatments for MS. The present work completes a cycle of preclinical testing (EAE → short-term testing in human PBMC cultures → long-term testing in antigen presentation cultures of DCs and T cells derived from human PBMCs) and demonstrates that the OM-MOG35–55 conjugate is the best candidate for human clinical trials to test whether it can be used as a therapeutic vaccine or as an immunomodulatory treatment for MS in the context of personalized medicine.

## 4. Materials and Methods

### 4.1. Study Subjects

Ten patients diagnosed with RRMS [1] and 10 healthy control subjects participated in the study (Table 1). All study participants donated peripheral blood, which was used for preliminary experiments to establish the experimental protocols described in this study. Additional blood samples from 5 RRMS patients and 5 controls from the original pool of patients and control subjects were used for the experiments presented in the results.

Ethics: This study was approved by the Scientific Review Boards and Ethics Committees of Patras University Hospital (Reg# 451/17.10.2008) and Eginition Hospital, National and Kapodistrian University of Athens, Athens, Greece (Reg# 560/30.07.2018) in the context of applications for studies on the pathogenesis of multiple sclerosis involving the use of clinical data from study participants (without disclosure of their names) and blood samples for in vitro experiments described in publications. Both hospitals adhere to the Declaration of Helsinki on the ethical principles of medical research involving human subjects.

### 4.2. Cells and Cultures

Peripheral blood samples (10–20 mL) were collected from RRMS patients and controls in heparinized BD vacutainers (Becton Dickinson, BD, Franklin Lakes, NJ, USA). The percentage of peripheral blood lymphocytes and monocytes in the blood samples is shown in Table 1. PBMCs were isolated via density gradient centrifugation with Ficoll (Biochrom, Feucht/Nuremberg, Germany) as described [32]. Cells were cultured in RPMI1640 medium (Gibco-BRL, Thermo Fisher Scientific Inc., Waltham, MA, USA) containing 10% fetal bovine serum (FBS, Gibco-BRL), 1% penicillin/streptomycin and 6 mM 2-mercaptoethanol (Sigma-Aldrich, St. Louis, MO, USA) (culture medium, CM) at a concentration of 2 × 10^6^ cells/mL for 2 h. At the end of the culture, non-adherent cells were collected and stored at 4 °C. 

For the differentiation of monocytes into different DC types, adherent cells (monocyte-enriched fraction) were cultured in CM containing GM-CSF and IL-4 (PeproTech, Thermo Fisher Scientific Inc.) at a concentration of 1000 IU/mL, as described in [33], in the presence or absence of 1 nM calcitriol (active form of vitamin D3, VITD3) or 10^−6^ M dexamethasone (DEXA) (Tocris Bioscience, Bristol, UK) or both for 6 days [20,21]. The CM with cytokines ± VITD3, DEXA or VITD3+DEXA was renewed every 2 days. At the end of their differentiation (day 6 of culture), the DCs were loaded with OM-MOG35–55 or MOG35–55 peptides that were added to the cultures at a concentration of 10 μg/mL. After 2 h, lipopolysaccharide (LPS, Sigma-Aldrich) was added to the cultures at a concentration of 0.25 μg/mL [22]. 

Two days later (day 8 of culture), the DCs were phenotyped using flow cytometry (see below) and co-cultured with non-adherent PBMCs that were added to the cultures together with IL-2 (PeproTech) at a concentration of 25 IU/mL. OM-MOG35–55 or MOG35–55 was added to the cultures at a concentration of 10 μg/mL every 11 days for a total of 4 antigen presentation cycles.

### 4.3. Flow Cytometry

Cells harvested from the cultures at different time points were analyzed on a BD FACSCanto™ II flow cytometer with fluorescently labeled monoclonal antibodies against CD3, CD4, CD8, CD14, CD25, CD40, CD45RA, CD45RO, CD56, CD69, CD80, CD83, CD86, CD279 (PD-1), HLA-DR and Foxp3 for phenotypic analysis (Table 2). Cell viability was determined using the PE Annexin V Apoptosis Detection Kit I (BD). At least 10,000 events were recorded for extracellular or intracellular staining. All measurements were performed in triplicate. Data analysis was performed using FlowJo V10.8 software (Tree Star Inc., San Carlos, CA, USA).

### 4.4. Measurement of Cytokines

The concentration of the cytokines IL-1β, IL-4, IL-6, IL-8, IL-10, IL-12p70, IL-17A, IFN-γ and TNF-α in the culture supernatant was measured using flow cytometry on a BD FACSArray™ Bioanalyzer using the CBA Human Inflammatory Cytokine Kit (BD) and the CBA Human Th1/Th2/Th17 Kit (BD). The concentration of TGF-β1 was measured using ELISA (R&D Systems, Minneapolis, MN, USA). All measurements were performed in triplicate. Data analysis was performed using FlowJo V10.8 software and GraphPad Prism 6.0 (GraphPad Software Inc., La Jolla, CA, USA).

### 4.5. Statistical Analysis

Data are presented as median or mean (SD) of at least three independent experiments. The Kolmogorov–Smirnov test was performed to determine distribution normality. Differences between two groups were analyzed using the unpaired Student’s *t*-test. A significance level of *p* < 0.05 was considered statistically significant. Data analysis and graphical representation were performed using GraphPad Prism 8.0 (GraphPad Software Inc.).

## Figures and Tables

**Figure 1 ijms-25-06092-f001:**
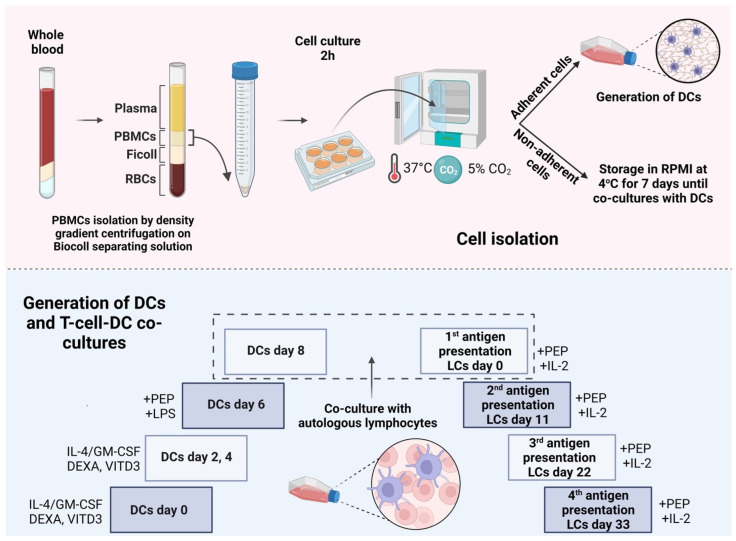
Protocol for in vitro differentiation of peripheral blood monocytes into different types of DCs presenting peptides to T cells. PEP, peptide; LCs, lymphocytes. This image was created with BioRender (https://biorender.com, accessed on 28 November 2023).

**Figure 2 ijms-25-06092-f002:**
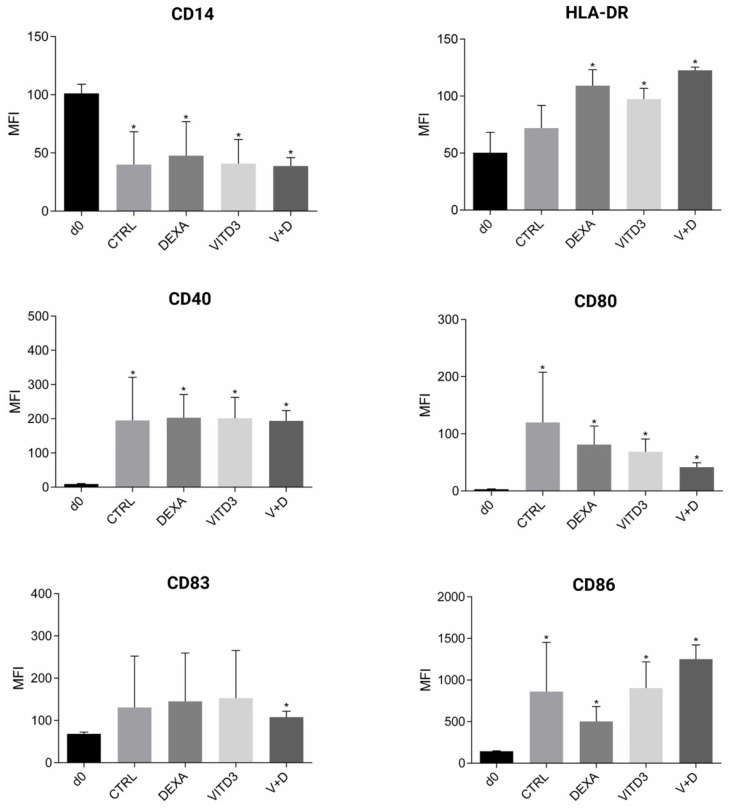
Phenotypic markers on the surface of cultured monocytes on day 0 and DCs on day 8. The results shown are from patient-derived cells. Data are presented as mean (SD). Asterisks indicate statistically significant differences between markers expressed on monocytes on day 0 and DCs on day 8 (* *p* < 0.05). MFI, mean fluorescence intensity; CTRL, control DCs; DEXA, DEXA-DCs; VITD3, VITD3-DCs; V+D, VITD3+DEXA; d0, day 0.

**Figure 3 ijms-25-06092-f003:**
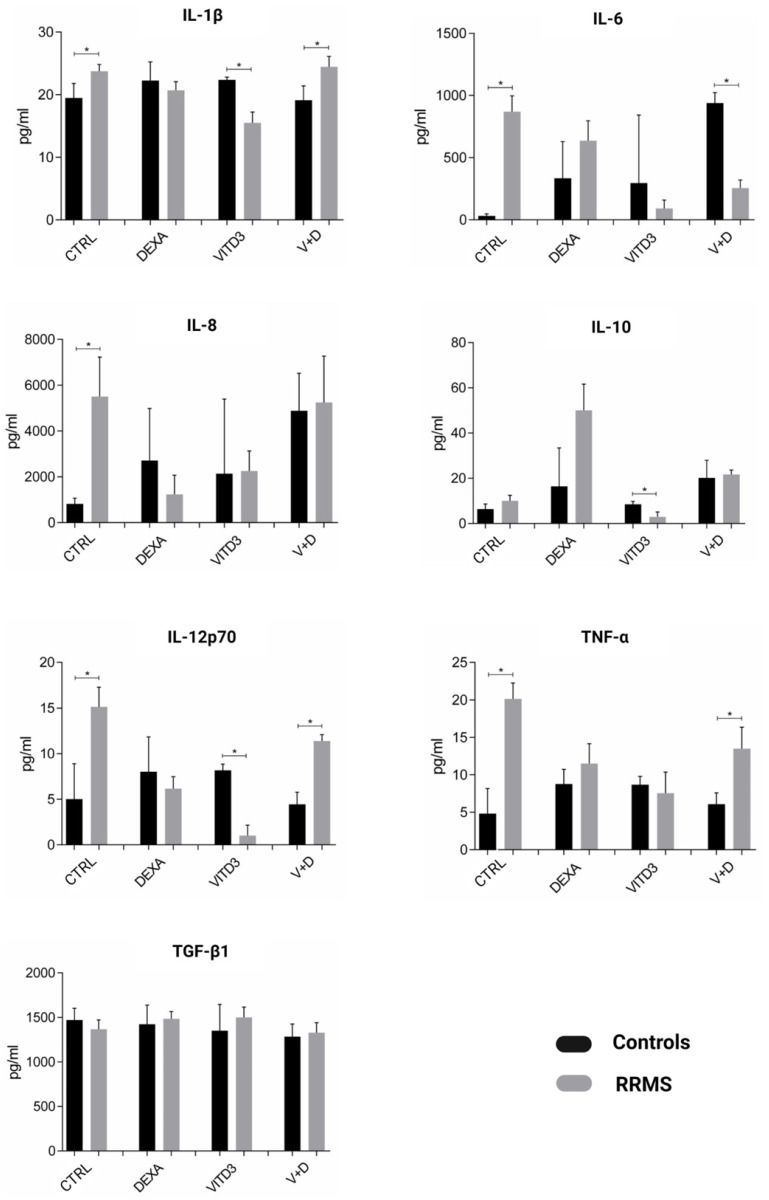
Cytokines secreted by DCs generated from controls or RRMS patients under different culture conditions on day 8 of culture. Data are presented as mean (SD). Asterisks indicate statistically significant differences between cytokine levels secreted by patient- and control-derived DCs (* *p* < 0.05).

**Figure 4 ijms-25-06092-f004:**
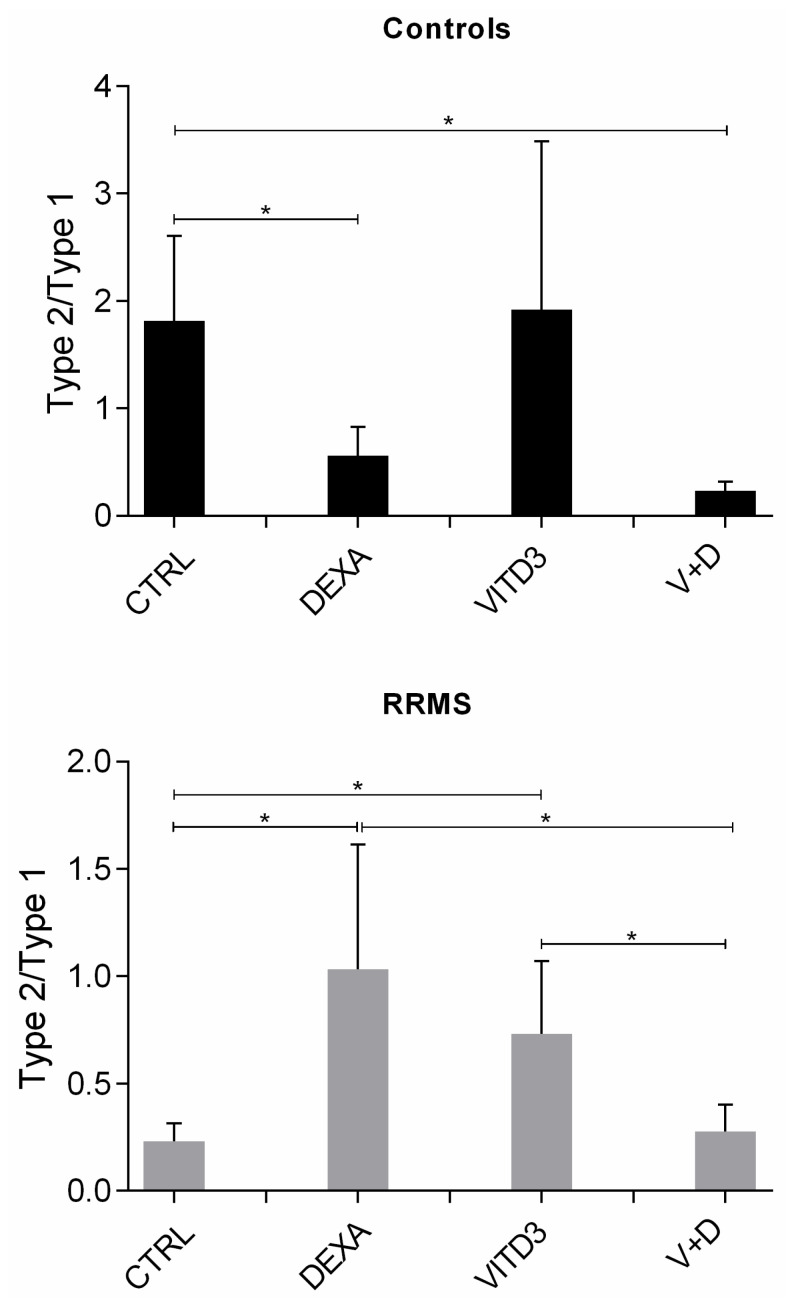
Ratio of type-2/type-1 cytokines in DC cultures on day 8. Data are presented as mean (SD). Asterisks indicate statistically significant differences between the ratios of type-2/type-1 cytokines (* *p* < 0.05). Type-2/type-1: [IL-10+TGF-β]:[IL-1+IL-6+IL-8+IL-12+TNF-α].

**Figure 5 ijms-25-06092-f005:**
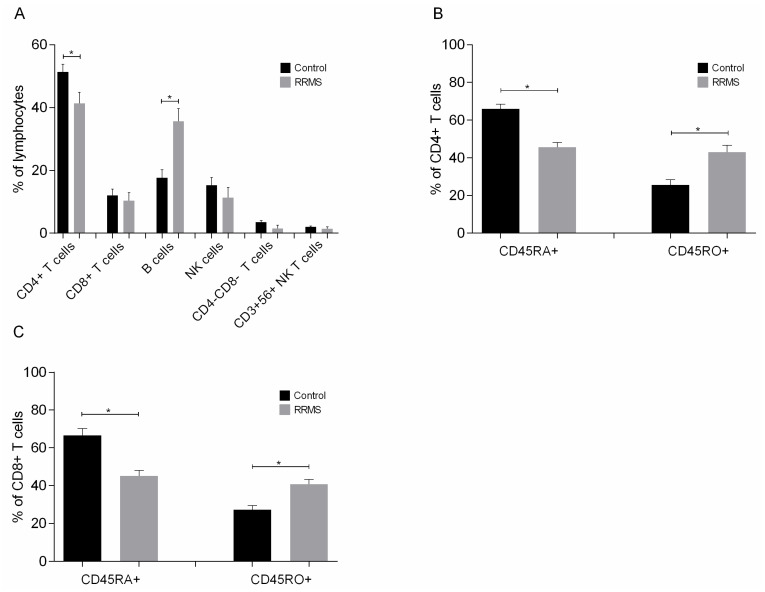
Phenotypic analysis of (**A**) lymphocytes, (**B**) naive and memory CD4+ T cells and (**C**) naive and memory CD8+ T cells added to DC cultures on day 8. Data are presented as mean (SD). Asterisks indicate statistically significant differences between cell concentrations in patients and controls (* *p* < 0.05). CD45RA+, naive cells; CD45RO+, memory cells.

**Figure 6 ijms-25-06092-f006:**
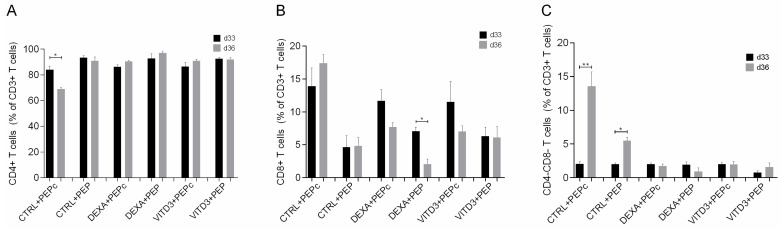
Phenotypic analysis of (**A**) CD4+ T cells, (**B**) CD8+ T cells and (**C**) CD4-CD8-T cells on days 33 and 36 of culture with DCs presenting the peptide OM-MOG35–55 (PEP) or the peptide MOG-35–55 (PEPc). Data are presented as mean (SD). Asterisks indicate statistically significant differences between cell levels (* *p* < 0.05, ** *p* < 0.01).

**Figure 7 ijms-25-06092-f007:**
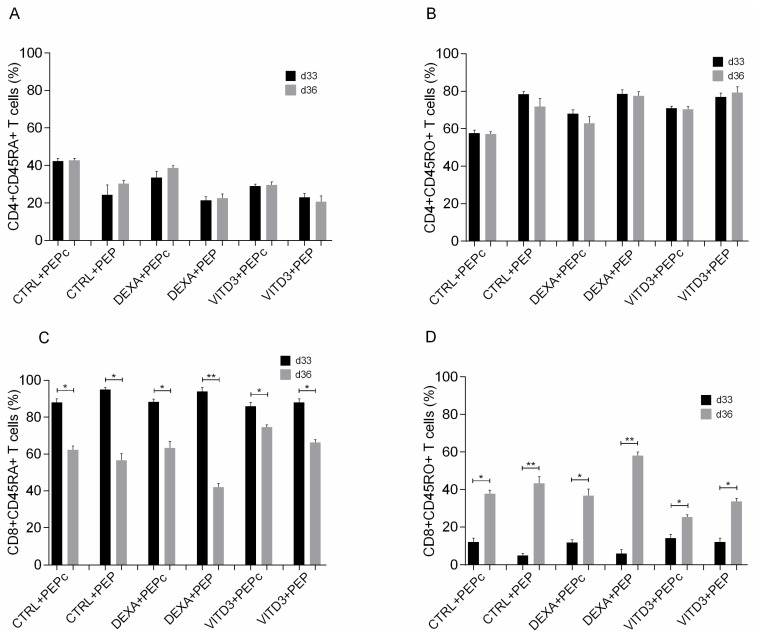
Phenotypic analysis of (**A**) naive CD4+ T cells, (**B**) memory CD4+ T cells, (**C**) naive CD8+ T cells and (**D**) memory CD8+ T cells on days 33 and 36 of culture with DCs presenting the peptide OM-MOG35–55 (PEP) or the peptide MOG-35–55 (PEPc). Data are presented as mean (SD). Asterisks indicate statistically significant differences between cell levels (* *p* < 0.05, ** *p* < 0.01). (%), data are presented as % of total CD4+ or CD8+ T cell populations.

**Figure 8 ijms-25-06092-f008:**
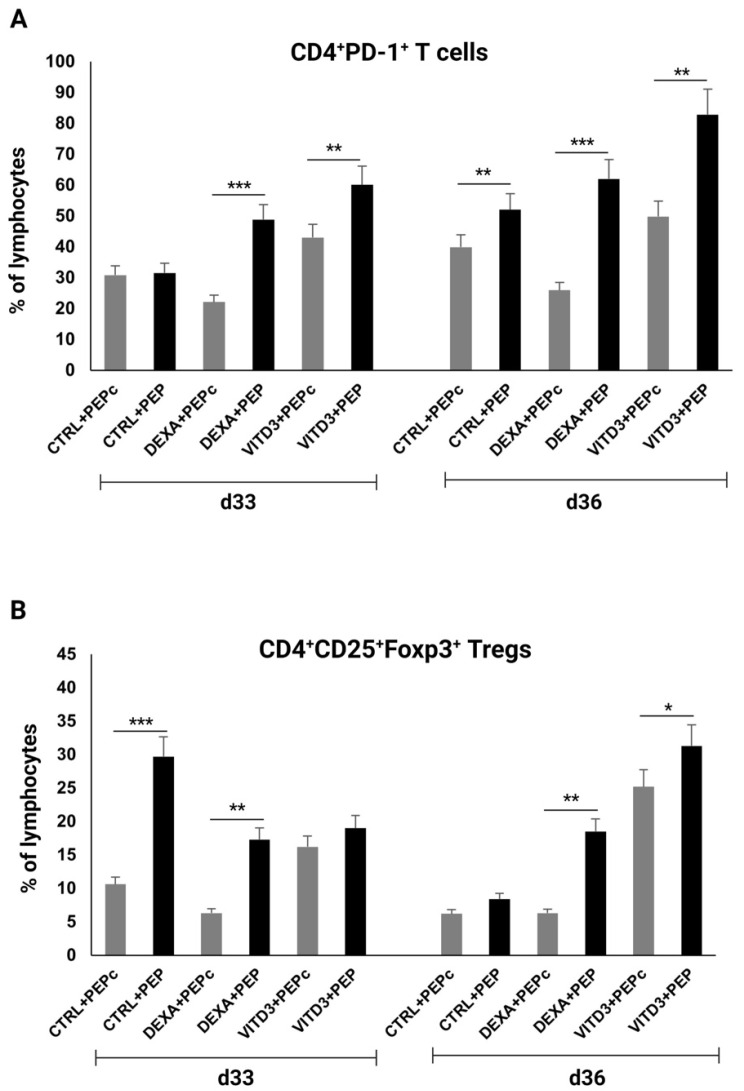
Percentage of (**A**) CD4+PD-1+ T cells and (**B**) CD4+CD25+Foxp3+ Tregs on days 33 and 36 of culture with DCs presenting peptide OM-MOG35–55 (PEP) or peptide MOG-35–55 (PEPc). Asterisks indicate statistically significant differences between CD4+PD-1+ T cell or Treg levels in cultures with control-derived T cells (gray bars) and patient-derived T cells (black bars). * *p* < 0.05; ** *p* < 0.01; *** *p* < 0.001.

**Figure 9 ijms-25-06092-f009:**
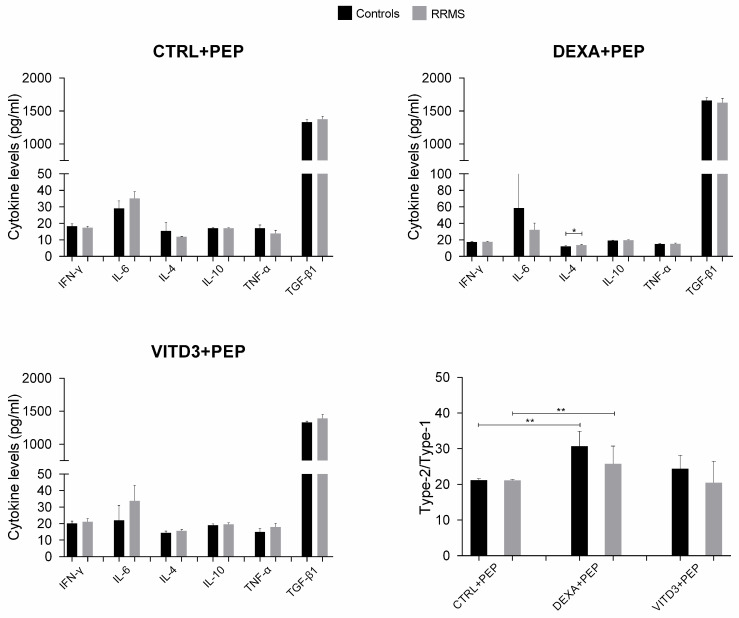
Cytokine concentration in supernatants of 36-day cultures of DCs and T cells derived from RRMS patients and controls. Cytokine concentrations are shown as mean (SD). Type-2/type-1 cytokine ratio: [IL-4+IL-10+TGF-β1]: [IFN-γ+IL-6+TNF-α]. * *p* < 0.05, ** *p* < 0.01. PEP, OM-MOG35–55.

**Figure 10 ijms-25-06092-f010:**
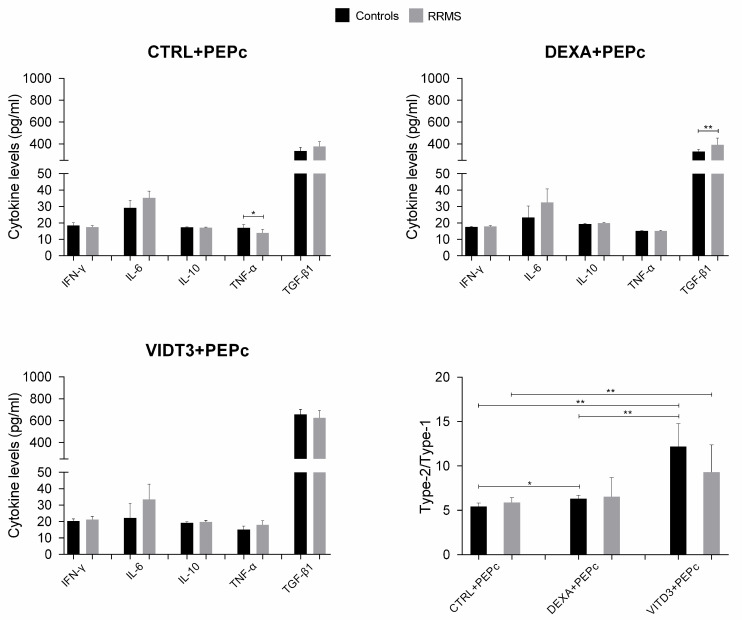
Cytokine concentration in supernatants of 36-day cultures of DCs and T cells derived from RRMS patients and controls. Cytokine concentrations are shown as mean (SD). Type-2/type-1 cytokine ratio: [IL-10+TGF-β1]: [IFN-γ+IL-6+TNF-α]. * *p* < 0.05, ** *p* < 0.01. PEPc, MOG35–55.

**Table 1 ijms-25-06092-t001:** Data of the study subjects.

Study Subjects	RRMS	Controls
*n* (M/F)	10 (4/6)	10 (5/5)
Age (range, y)	24–38	24–38
Disease duration (range, y)	2–7	na
EDSS (range)	1–2.5	na
Treatment with IFNβ (*n*)	6	na
% of lymphocytes in peripheral blood (mean (SD))	35.19 (9.14)	28.97 (10.19)
% of monocytes in peripheral blood (mean (SD))	7.25 (1.53)	6.86 (1.49)

RRMS patients: Medical history and neurological examinations were used to assess disease progression, current status, disease duration and degree of disability according to the Expanded Disability Status Scale (EDSS) [31]. *n*, number of subjects; M, male; F, female; y, years; na, not applicable; SD, standard deviation.

**Table 2 ijms-25-06092-t002:** Antibodies used for phenotypic analysis.

Target Antigen	Company	Clone	Fluorochrome
CD3	Beckman Coulter	UCHT1	PC5
CD4	Beckman Coulter	13B8.2	FITC
CD8	Beckman Coulter	B9.11	FITC
CD8	Becton Dickinson	PRA-T8	PE
CD14	Becton Dickinson	M5E2	FITC
CD25	Becton Dickinson	M-A251	PE
CD40	Becton Dickinson	5C3	PE
CD45-RA	Becton Dickinson	HI100	PE
CD45-RO	Becton Dickinson	UCHL1	PE
CD56	Beckman Coulter	N901	PE
CD69	Becton Dickinson	FN50	PE
CD80	Becton Dickinson	L307.4	PE
CD83	Becton Dickinson	HB15e	PC5
CD86	Becton Dickinson	IT2.2	PE
CD279 (PD-1)	Beckman Coulter	PD1.3	PE
HLA-DR	Becton Dickinson	TU36	PE
HLA-DR	Becton Dickinson	TU36	PC5
Foxp3	eBioscience	PCH101	PC5

Note: Beckman Coulter, Brea, CA, USA; Becton Dickinson, Franklin Lakes, NJ, USA; eBioscience, San Diego, CA, USA.

## Data Availability

Data are contained within the article.

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
