# Peer review of "Myelin Oligodendrocyte Glycoprotein (MOG)35–55 Mannan Conjugate Induces Human T-Cell Tolerance and Can Be Used as a Personalized Therapy for Multiple Sclerosis"

_ijms, 2024, doi:10.3390/ijms25116092_

Round 1
Reviewer 1 Report (New Reviewer)
Comments and Suggestions for Authors
The manuscript that you submitted is of high quality. The introduction provide the rational and the necessary background. The results are clear and not misleading. The materials and methods section describes well the experiments.
I only have few minor comments:
Introduction: I would suggest to clearly state the hypothesis.
Material and Methods: It's not clear for me how you differentiated the monocyte isolated from PBMCs into DCs. I would suggest to develop the processes in this section
In figure 2 you analyzed CD14, did you take a look at the CD16+ population? It would be interesting to see which monocyte subpopulation is differentiated into DC.
The discussion could be improved by discussing the limitations of this study and what would be the next steps.
Thanks
Author Response
Re: ijms- 3004121-R1
Myelin oligodendrocyte glycoprotein (MOG)35-55 mannan conjugate induces human T-cell tolerance and can be used as a personalized therapy for multiple sclerosis
Responses to Reviewers
Taking into account the reviewers’ comments on the revised manuscript (Re: ijms-2303261-R2), we have made a third revision that presents our methods, results and discussion more clearly, including additional references.
As requested by the editor, we have also changed the format of our manuscript and reference list to comply with journal regulations.
All changes/additions in the revised manuscript are highlighted in yellow.
Responses to the specific comments of Reviewer 1
Introduction: I would suggest to clearly state the hypothesis.
Our working hypothesis is that the OM-MOG35-55 conjugate is a strong candidate for a therapeutic vaccine or immunomodulatory treatment of MS in the context of personalized medicine. Added in the Introduction (cf. lines 95-97).
Material and Methods: It's not clear for me how you differentiated the monocyte isolated from PBMCs into DCs. I would suggest to develop the processes in this section
The experimental procedure for the differentiation of monocytes isolated from PBMCs into DCs is described in more detail in the M&M - Cells and cultures (cf. lines 310-323).
In figure 2 you analyzed CD14, did you take a look at the CD16+ population? It would be interesting to see which monocyte subpopulation is differentiated into DC.
You raise an interesting point and we will follow this suggestion in our future experiments. The characterization of monocyte subsets in the course of maturation was beyond the scope of this study, and the flow cytometry panels were designed with a focus on the co-expression of monocyte (CD14) and DC maturation antigens. However, there is some interesting research (mainly related to malignancy, where the relative proportions of monocyte subsets are altered) that points to differences in the phenotype and activation potential of DC generated by CD16+ (intermediate and non-classical) and CD16- classical monocytes in humans (Sánchez-Torres et al. 2001; Randolph et al. 2002) and different subsets in mice (Auffray et al. 2009; Geissmann et al. 2010).
References
Auffray C, Sieweke MH, Geissmann F. Blood monocytes: development, heterogeneity, and relationship with dendritic cells. Annu Rev Immunol. 2009; 27:669-692. doi: 10.1146/annurev.immunol.021908.132557.
Geissmann F, Manz MG, Jung S, Sieweke MH, Merad M, Ley K. Development of monocytes, macrophages, and dendritic cells. Science. 2010; 327(5966):656-661. doi: 10.1126/science.1178331.
Randolph GJ, Sanchez-Schmitz G, Liebman RM, Schäkel K. The CD16(+) (FcgammaRIII(+)) subset of human monocytes preferentially becomes migratory dendritic cells in a model tissue setting. J Exp Med. 2002; 196(4):517-527. doi: 10.1084/jem.20011608.
Sánchez-Torres C, García-Romo GS, Cornejo-Cortés MA, Rivas-Carvalho A, Sánchez-Schmitz G. CD16+ and CD16- human blood monocyte subsets differentiate in vitro to dendritic cells with different abilities to stimulate CD4+ T cells. Int Immunol. 2001; 13(12):1571-1581. doi: 10.1093/intimm/13.12.1571.
The discussion could be improved by discussing the limitations of this study and what would be the next steps.
The limitations of this study and our plans for the next steps in this project are discussed (cf. lines 250-258).
Reviewer 2 Report (New Reviewer)
Comments and Suggestions for Authors
The manuscript entitled “Myelin Oligodendrocyte Glycoprotein (MOG)35–55 Mannan Conjugate Induces Human T-cell Tolerance and can be used as a Personalized Therapy for Multiple Sclerosis” describe the results from the performed study in which developed dendritic cells (DCs) from peripheral blood monocytes of multiple sclerosis (MS) patients and from healthy subjects were manipulated in vitro to induce and maintain T cell tolerance.
The study is based on previous in vivo (EAE mouse model) and in vitro (human peripheral blood) studies performed by authors in which they demonstrated that the oxidized mannan-conjugated peptide MOG35–55 (OM-MOG35–55) suppresses antigen-specific T cell responses associated with autoimmune demyelination.
Overall the manuscript is well written using scientifically sound english and the experimental design is appropriate. I have just some minor comments on it:
- Figure 2: It is not clear what experimental group (RRMS or healthy subjects) presented results come from. It should be specified.
In this figure, results are presented as Mean Fluorescent Intensity. Why are they not quantified?
- In figures, authors use different colors to represent the same experimental group, e.i. in figures 3, 4, 5 and 8 controls are black and RRMS are grey, but in figure 9 and 10 controls are grey and RRMS are purple. This could be a little bit confusing.
- Pag 14, line 314, the abbreviation TCR should be specified.
Author Response
Re: ijms- 3004121-R1
Myelin oligodendrocyte glycoprotein (MOG)35-55 mannan conjugate induces human T-cell tolerance and can be used as a personalized therapy for multiple sclerosis
Responses to Reviewers
Taking into account the reviewers’ comments on the revised manuscript (Re: ijms-2303261-R2), we have made a third revision that presents our methods, results and discussion more clearly, including additional references.
As requested by the editor, we have also changed the format of our manuscript and reference list to comply with journal regulations.
All changes/additions in the revised manuscript are highlighted in yellow.
Responses to the specific comments of Reviewer 2
Figure 2: It is not clear what experimental group (RRMS or healthy subjects) presented results come from. It should be specified.
The results shown in Figure 2 are from the RRMS group, but as mentioned in the Results (cf. lines 116-117), the results between patient and control monocytes and DCs were (very) similar, and for this reason showing both groups would only clutter the figure. We have indicated in the figure legend that the results shown are from patient-derived cells (cf. lines 119-120).
In this figure, results are presented as Mean Fluorescent Intensity. Why are they not quantified?
The expression of surface molecules such as CD14, CD40, HLA-DR, CD80, CD83 and CD86 on monocytes and DCs follows a pattern of continuous expression with variable levels depending on their degree of maturation. Monocytes express mainly CD14 and lower levels of HLA-DR and CD83, immature DCs express less CD14 and lower levels of CD80, CD83 and CD86, and mature DCs (or tolerogenic semi-mature DCs) upregulate these markers to varying degrees (Mellman and Steinman 2001; Hubo et al. 2013). The nature of this process is reflected in the different mean fluorescence intensity of each marker in flow cytometric analysis, rather than the conventional clear clustering of “positive” and “negative” populations. Furthermore, although the MFI correlates with the number of antibodies that recognize and attach to the cell antigens, it cannot be quantified to an actual measurement of the molecules on the cell surface due to a lack of standardized references (Mizrahi et al. 2018).
References
Hubo M, Trinschek B, Kryczanowsky F, Tuettenberg A, Steinbrink K, Jonuleit H. Costimulatory molecules on immunogenic versus tolerogenic human dendritic cells. Front Immunol. 2013; 4:82. doi: 10.3389/fimmu.2013.00082. (Ref. 23 in the manuscript)
Mellman I, Steinman RM. Dendritic cells: specialized and regulated antigen processing machines. Cell. 2001; 106(3):255-258. doi: 10.1016/s0092-8674(01)00449-4. (Ref. 24 in the manuscript)
Mizrahi O, Ish Shalom E, Baniyash M, Klieger Y. Quantitative Flow Cytometry: Concerns and Recommendations in Clinic and Research. Cytometry B Clin Cytom. 2018; 94(2):211-218. doi: 10.1002/cyto.b.21515.
In figures, authors use different colors to represent the same experimental group, e.i. in figures 3, 4, 5 and 8 controls are black and RRMS are grey, but in figure 9 and 10 controls are grey and RRMS are purple. This could be a little bit confusing.
Figures 9 and 10 have been revised accordingly (cf. Figures 9 and 10).
Pag 14, line 314, the abbreviation TCR should be specified.
Done in the Introduction, where it appears for the first time (cf. line 60).
Reviewer 3 Report (New Reviewer)
Comments and Suggestions for Authors
In the current study, the authors examined the tolerogenic potential of OM-MOG35–55 for its possible use in MS therapy
They show that the OM-MOG35–55 conjugate is the most appropriate choice for human clinical trials to investigate its potential applications as an immunomodulatory treatment for MS or as a therapeutic vaccination.
The results are interesting. The figures reflect the results of the study. However, there are some concerns that need to be addressed.
Which tests were used to test a data set for normality?
In cells and culture region, please add the references for PBMCs isolation by density gradient centrifugation with Ficoll.
Regarding the measure of TGF-β1 concentration, what was the variation between samples in this assay?
The references for concentration of vitamin D3,(VITD3) and 10−6 M dexamethasone (DEXA) should be written in the text.
Author Response
Re: ijms- 3004121-R1
Myelin oligodendrocyte glycoprotein (MOG)35-55 mannan conjugate induces human T-cell tolerance and can be used as a personalized therapy for multiple sclerosis
Responses to Reviewers
Taking into account the reviewers’ comments on the revised manuscript (Re: ijms-2303261-R2), we have made a third revision that presents our methods, results and discussion more clearly, including additional references.
As requested by the editor, we have also changed the format of our manuscript and reference list to comply with journal regulations.
All changes/additions in the revised manuscript are highlighted in yellow.
Responses to the specific comments of Reviewer 3
Which tests were used to test a data set for normality?
The Kolmogorov-Smirnov test was performed to determine distribution normality. Added in M&M - Statistical analysis (cf. lines 345-346).
In cells and culture region, please add the references for PBMCs isolation by density gradient centrifugation with Ficoll.
Added in M&M - Cells and cultures (cf. line 305 and ref. 32).
Regarding the measure of TGF-β1 concentration, what was the variation between samples in this assay?
If you mean the sample variance in terms of measurement of each sample in triplicate to get a mean value/sample, then the variance was very low. For example, for a sample that gave a mean concentration of TGF-β=1608 ng/ml, the variance (s2) was 7.
If you mean the sample variance in terms of measurement of the (different) samples derived from different patients or controls/same experimental point, this was done by measuring the SD.
The references for concentration of vitamin D3,(VITD3) and 10−6 M dexamethasone (DEXA) should be written in the text.
They are written in the text in Results - Development of DCs from peripheral blood monocytes (cf. lines 102-103) and in M&M - Cells and cultures (cf. lines 312-314); Refs 20, 21.
This manuscript is a resubmission of an earlier submission. The following is a list of the peer review reports and author responses from that submission.
Round 1
Reviewer 1 Report
Comments and Suggestions for Authors
The manner in which the paper is presented it is difficult to determine whether it is intended as a review or research paper. If it is a research paper section 2 is not entirely appropriate and should be briefly summarized in the introduction merely to inform the reader about the nature of the study. Limitations of previous experiments should also be clear, for example that many of the studies described were evidently performed in a single mouse model and peptide recognition is likely to differ between individuals. The data presented lacks detail including variation and statistical analysis. Moreover conclusions are drawn from phenotypic but minimal functional data. Without showing the levels of cytokines produced by the different cultures it is impossible to even use this data for interpretation. Finally, while the limited data raises some interest, elaboration of the role of vitamin D3 and dexamethasone in the induction of putatively "tolerogenic" DC and actual CD80 and CD86 values should be provided.
Author Response
Re: ijms-2303261-R1
Myelin oligodendrocyte glycoprotein (MOG)35-55 mannan conjugate induces tolerogenic T cells: Immune modulation in multiple sclerosis (MS)
Responses to Reviewers
Taking into account the comment made by both reviewers on the original submission, namely that it was not clear whether it was a review article or a research paper, and taking into account the requests of the Managing and Assistant Editors, we have completely changed the paper to be a typical research paper with the formatting required by the journal.
In addition, we have changed the abstract, shortened the introduction, and expanded the Methods, Results, and Discussion sections to focus on the novel research we present in this manuscript (hence the small change in title).
We ask the reviewers to read the revised manuscript, as we believe that all the points they raised have been met.
We also ask the reviewers to consider that the research presented in this paper has taken a lot of time and effort and, in our opinion, will stimulate further research on the topic of personalized treatment of patients with multiple sclerosis (or other autoimmune diseases).
Reviewer 2 Report
Comments and Suggestions for Authors
Abstract.
The abstract does not reflect the content of the article, but contains a lot of historical information - must be changed.
Introduction.
Currently available drugs in the EU alone include many more preparations than those enumerated in section 2.1 However, since this is an experimental study, perhaps it is not necessary to go into a detailed description of currently approved preparations? Likewise, what is the purpose of reviewing experimental studies with MBP protein?
The introduction is too long and contains a great deal of information not at all related to the study(which we learn about on the fifth page of the macnuscript! Until then, I thought I was dealing with a literature review.
For example, is the entire section 2.4 needed, since MBP83-99 is not used later in the study?
AIM of the study
Missing - clearly stated aim of the study. It is only from the conclusions that one learns that the authors want to present development of a new antigen presentation/delivery system
Author Response

(The authors gave the same response as above.)

Round 2
Reviewer 1 Report
Comments and Suggestions for Authors
This is an interesting study that suggests the therapeutic potential for DC differentiated to a tolerogenic form. However, there are a number of concerns with the manuscript.
1. It appears that the patient cells studied were from only 2 individuals which is not sufficient to support substantive conclusions.
2. The data presented is absent of error bars such that the significance of any data presented cannot be judged.
3. Much of the T cell data presented is phenotypic and descriptive without functional information.
4. A number of subsets of tolerogenic APC with DC have been identified with specific phenotypic and functional attributes (Thetis cells). More detailed analysis of the APC that present OM-MOG35-55 in the current study is critical to understanding their genesis.
5. Some of the data presented, for example viability flow cytometry, should be noted but is not substantive enough to have a dedicated figure.
Author Response
Re: ijms-2303261-R2
Myelin oligodendrocyte glycoprotein (MOG)35-55 mannan conjugate induces human T-cell tolerance and can be used as a personalized therapy for multiple sclerosis
Responses to Reviewer 1
Taking into account the comments of Reviewer 1 on the revised manuscript (ijms-2303261-R1), we have made a second revision that includes results from more patients and healthy controls and presents our results more clearly, including the statistical analysis. For comparison purposes, we have also included results from additional experiments with the wild-type peptide MOG35-55.
Although the main results of our study have not changed, we believe that the present manuscript better conveys our findings from a research project that took a lot of time and effort and demonstrates that the oxidized mannan-conjugated peptide MOG35-55 developed by our group can form the basis for the development of personalized therapeutic vaccines or immunomodulatory treatments for MS.
For these reasons, we have changed the title and abstract, shortened the introduction, expanded the methods and results sections, and changed the discussion to focus on the novel research we present in this manuscript. All changes/additions in the revised manuscript are highlighted in yellow.
Responses to the specific comments of Reviewer 1
- It appears that the patient cells studied were from only 2 individuals which is not sufficient to support substantive conclusions.
The revised manuscript contains results from more patients and healthy control subjects (cf. section 2.1. Study subjects, lines 110-128).
- The data presented is absent of error bars such that the significance of any data presented cannot be judged.
In the revised manuscript, we have presented the data with statistical analysis and error bars (cf. section 2.5. Statistical analysis, lines 168-172; Figures 2-10; Supplementary Figure S1).
- Much of the T cell data presented is phenotypic and descriptive without functional information.
Function is depicted by measuring the cytokines secreted in the culture supernatants at the end of the 36-day culture and after four rounds of antigen presentation (cf. section 3.3.2.2. Cytokines, lines 283-296; Figures 9 and 10). In addition, the observed changes in T cell phenotypes and in particular the significant increase in CD4+PD-1 + T cells and Tregs when cultured with VITD3-DCs underline the response of T cells to the peptides (cf. section 3.3.2.1. Cells, lines 242-281; Figures 6-8; Discussion, lines 320-331).
- A number of subsets of tolerogenic APC with DC have been identified with specific phenotypic and functional attributes (Thetis cells). More detailed analysis of the APC that present OM-MOG35-55 in the current study is critical to understanding their genesis.
Details of the different types of dendritic cells generated from peripheral blood monocytes under different culture conditions can be found in section 2.2. Cells and cultures, lines 130-147; Results, sections 3.1. Development of DCs from peripheral blood monocytes and 3.2. Cytokines secreted by the different DC types, lines 175-222; Figures 1-4. The wild-type peptide MOG35-55 was also included in all experiments and results for comparison purposes (see also Discussion, lines 311-317).
- Some of the data presented, for example viability flow cytometry, should be noted but is not substantive enough to have a dedicated figure.
In the revised manuscript, we have moved the figure showing the viability of lymphocytes in stored PBMCs to the Supplementary Material (cf. Supplementary Figure S1).